# Biological Activated Sludge from Wastewater Treatment Plant before and during the COVID-19 Pandemic

**DOI:** 10.3390/ijerph191811323

**Published:** 2022-09-08

**Authors:** Marius-Daniel Roman, Cornel Sava, Dana-Adriana Iluțiu-Varvara, Roxana Mare, Lavinia-Lorena Pruteanu, Elena Maria Pică, Lorentz Jäntschi

**Affiliations:** 1Faculty of Building Services Engineering, Technical University of Cluj-Napoca, 28 Memorandumului Street, 400114 Cluj-Napoca, Romania; 2Faculty of Engineering Materials and the Environment, Technical University of Cluj-Napoca, 28 Memorandumului Street, 400114 Cluj-Napoca, Romania; 3Department of Chemistry and Biology, North University Center at Baia Mare, Technical University of Cluj-Napoca, 76 Victoriei Street, 430122 Baia Mare, Romania; 4Department of Physics and Chemistry, Technical University of Cluj-Napoca, 103-105 Bd. Muncii, 400641 Cluj-Napoca, Romania; 5Institute for Doctoral Studies, Babes-Bolyai University, 1 M. Kogălniceanu Street, 400084 Cluj-Napoca, Romania

**Keywords:** biological activated sludge, wastewater treatment plant, COVID-19 pandemic, microorganisms, bacteria, protozoa, rotifers, nitrification tank, microscopy

## Abstract

The COVID-19 pandemic and the related measures brought a change in daily life that affected the characteristics of the municipal wastewater and further, of the biological activated sludge. The activated sludge process is the most widely used biological wastewater treatment process in developed areas. In this paper, we aim to show the situation of specific investigations concerning the variation of the physicochemical parameters and biological composition of the activated sludge from one conventional wastewater treatment plant from a metropolitan area. The investigations were carried out for three years: 2019, 2020 and 2021. The results showed the most representative taxa of microorganisms: *Microtrix*, *Aspidisca cicada*, *Vorticella convallaria*, *Ciliata* free of the unknown and *Epistylis* and *Rotifers*. Even if other microorganisms were found in the sludge flocs, their small presence did not influence in any way the quality of the activated sludge and of the wastewater treatment process. That is why we conclude that protozoa (especially *Flagellates* and *Ciliates*) and rotifers were the most important. Together with the values and variation of the physicochemical parameters, they indicated a good, healthy, and stable activated sludge, along with an efficient purifying treatment process, no matter the loading conditions.

## 1. Introduction

COVID-19 started to affect the entire world at the beginning of 2020 and this was manifested further at all levels. The effects of the coronavirus lockdown were felt all over the globe, no matter the sector: industry, research, teaching, commerce, tourism, etc. [1]. All workplaces shut off for a shorter or longer period depending on the COVID-19 incidence and/or the precautionary measures proposed to reduce or eliminate exposure. People started to work from home or entered a phase of technical unemployment. All these aspects are later reflected in water consumption and the quality of the wastewater further processed and treated by the wastewater treatment plants (WWTPs). During the wastewater treatment process, the biological phase is extremely important as it shows the efficiency of the WWTP under various loading conditions [2]. The efficiency is further translated into the biological composition of the activated sludge. Therefore, questions arise regarding the influence of the COVID-19 pandemic and lockdown on the change of the physicochemical parameters and biological activated sludge, and its significance, respectively.

According to the literature, various studies focused on wastewater and activated sludge with respect to the fate of SARS-CoV-2 [3,4,5,6]. The first step was tracking [7] and surveillance [8] of COVID-19 on wastewater by analyzing samples of raw sewage and activated sludge to determine the RNA concentration. Once detecting the SARS-CoV-2 RNA in WWTPs, the removal part followed; from this point, ref. [9] demonstrated that the conventional activated sludge can be a risk for WWTP workers.

However, wastewater treatment plants in COVID-19 times do not represent only these directions regarding the fate of the disease. Due to the spread of prevention actions, temporary lockdown, stay-at-home policies, and teleworking, the characteristics of municipal wastewater changed. This is reflected in the physicochemical parameters and biological composition of the activated sludge, and further in the performance of WWTPs. The performance of the WWTPs seems to remain rather constant, even though the inflow increased while Chemical Oxygen Demand (COD) and Biochemical Oxygen Demand (BOD_5_) decreased [10,11]. With all of these, there is no information regarding the change or no change in the composition of the activated sludge from the biological mean and its correlation to the physicochemical parameters.

Therefore, the aim of this study is to evaluate and present: (1) the variation of the physicochemical parameters and (2) the microscopic composition of the biological activated sludge from a municipal wastewater treatment plant situated in Romania, in a metropolitan area, before the COVID-19 pandemic compared to the modifications brought by the COVID-19 pandemic and all the related measures. The reasons why we have chosen this study are: (i) the lack of this type of information in the literature and (ii) the lack of this comparison in the area and further in Romania. The authors’ intention was to reveal and demonstrate whether the COVID-19 pandemic changed or not the physicochemical parameters and biological composition of the activated sludge, and if yes, in what way and how.

Activated sludge can be defined as a microbial community and consists of biological flocs that are matrices of microorganisms, non-living organic matter, and inorganic materials. The species of these communities represent an important indicator of the wastewater treatment process efficiency. The microorganisms include bacteria, unicellular, fungi, protozoa, rotifers, insect larvae, and worms of different types [12]. An example of investigation regarding the microscopic determination was depicted in Ghana Wastewater Treatment Plant on filamentous bacteria, involving the application of morphological and molecular techniques [13]. The activated sludge process can also be defined as a system in which biological flocs are continuously circulated to meet the organic substances and oxidize them, in the presence of oxygen. As is well known, in the activated sludge process, the principal role of microorganisms is to convert dissolved and particulate organic matter (measured as BOD_5_) into cell mass [14] that will decrease with the Endogenous Phase [15,16].

When the activated sludge system is first started up and the activated sludge is very young and thin, *Amoeba* and some *Flagellates* are seen under the microscope. Once the sludge age increases, *Flagellates* and *free-swimmers ciliates* appear. When the activated sludge age reaches the optimum level for treatment, *Flagellates*, *free-swimming,* and *stalked ciliates* are observed to be more abundant [17,18,19].

Besides these, both nitrifying and denitrifying bacteria are present in large numbers in the activated sludge process. Nitrifying bacteria were derived from the carbon and energy substrates when autotrophic and heterotrophic nitrifiers were present in the activated sludge [20,21,22,23,24]. Denitrifying bacteria require the presence of substrate BOD_5_ and an adequate pH, temperature, nutrients, and redox potential [25,26]—the reason is that most denitrifying bacteria cannot ferment and use a molecule of carbonaceous biochemical oxygen demand (BOD_5_) to degrade another molecule of BOD_5_ as described in [27,28,29,30], whose level will be further established for our WWTP.

Therefore, according to all stated above, the goal of this paper is to emphasize if the COVID-19 pandemic influenced or not the variation of the physicochemical parameters and microscopic composition of the activated sludge when referring to the following microorganisms: bacteria, unicellular sessile, free unicellular (further presented as unicellular), and pluricellular that were analyzed. For this purpose, samples of the activated sludge from the aeration tanks of a WWTP situated in a metropolitan area of Romania were considered and analyzed before and during the COVID-19 times (years: 2019 before COVID-19, respectively 2020 and 2021 during COVID-19).

## 2. Materials and Methods

### 2.1. Site Description and Sampling

The whole study was conducted in a big conventional WWTP situated in a metropolitan area in Romania. As in a conventional activated sludge plant, the primary-treated wastewater and acclimated microorganisms (activated sludge or biomass) were aerated in tanks. After a sufficient aeration period, the flocculent activated sludge solids were separated from the wastewater in a secondary clarifier. The clarified wastewater flew forward for further treatment or discharge, according to the indications given by Guyer in 2013 [31].

We considered for determinations, two aeration tanks of the biological treatment with the following dimensions: width of 16.5 m, height of 5.5 m, and length of 89.6 m, having a volume of 8131 m^3^. The objective of the examination is to determine the relative predominance of the microorganisms in the specific tanks. The study took place over three years (2019, 2020, and 2021), considering a better and proper comparison of the biological activated sludge before and during the COVID-19 pandemic.

March 2020 represented the start of the Romanian lockdown [32]. In 2021, the economy has regained, which means activities for the industry, commerce, services, etc., translated into changes in the characteristics of the wastewater and activated sludge. To observe these changes, microscopic analyses of the wastewater and activated sludge were performed. In this way, the wastewater samples had been manually collected in plastic containers, directly from the aeration tanks. In the laboratory, the wastewater samples were poured into Erlenmeyer flasks and further put into slides using single-use Pasteur pipettes. For the biological microscopic characterizations, 118 total samples of activated sludge were taken from the WWTP aeration tanks, in the years 2019, 2020, and 2021. In 2019, 74 samples were taken (37 samples from aeration tank 1 and 37 samples from aeration tank 2). In 2020, 30 samples were taken (15 samples from aeration tank 1 and 15 samples from aeration tank 2). In 2021, 14 samples were taken (7 samples from aeration tank 1 and 7 samples from aeration tank 2). Microscopic characterization was performed on average monthly samples.

In most cases of research in the biological field, the most important characteristic is given by the number and frequency of the various microorganisms that dominate the activated sludge in the aeration tanks. Their preparation and evaluation must always be based on data from measured samples. Thus, the number of microorganisms can be observed at the microscope, and their number is transformed into notes of abundance.

### 2.2. Determination of Physicochemical Parameters

In Romania, the mandatory parameters for big WWTPs after primary sedimentation tanks are the following: pH, BOD_5_, COD, TSS, TN, TP, all in mg/L per day. The assessment of various physicochemical parameters was carried out as per the method: pH was determined by SR ISO 10523: 2012 [33]; chemical oxygen demand (COD) by SR ISO 6060: 1996 [34]; biochemical oxygen demand after 5 days (BOD_5_) according to SR EN 5815-1: 2020. Part 1 [35] and SR EN 1899-2: 2002. Part 2 [36]; suspended solids (TSS)—Method by filtration through glass fiber filters SR EN 872: 2005 [37]; nitrogen (TN) by SR EN ISO 11905–1: 2003. Part 1 [38], phosphorus (TP) using ammonium molybdate according to SR EN ISO 6878: 2005 Chapter 7 [39].

### 2.3. Microscopic Analyses

The microscopic observations at municipal wastewater treatment plants made possible a more complete and defined analysis.

In order to microscopically characterize the activated sludge samples taken from the aeration tanks, a trinocular microscope of the Leica DM2000 type was used, which is equipped with 15× WF eyepieces; variable lighting; capable of light field and dark field observations; capable of phase contrast observations; equipped with a turret with five apochromatic objectives: 5×, 10×, 20×, 40×, 100×; equipped with real 5 Mpx camera (not interpolated), software and computer connection; filter holder (green filter and blue filter). A schematic presentation of the microscopic characterization of the activated sludge can be seen in Figure 1.

Microscopic determination of the mixed liquor suspended solids (MLSS) can bring a significant benefit in the evaluation of the activated sludge process. Unfortunately, the heterotrophic and autotrophic bacteria, which are firstly responsible for the purification of wastewater, are too small to be easily observed. The presence of several other microorganisms within the sludge flocs may also help form a clear image of the treatment conditions and efficiency. The most important of these microorganism indicators is the presence of *protozoa* and *rotifers*. These higher life forms also play an important role in clarifying the wastewater, consuming bacteria and small particles, and improving biomass settleability [40].

Tracking cells through microscopic images are suggestive when continuously done regularly and systematically. For a good observation, at least 3 microscopic preparations of the activated sludge sample are analyzed, and an average of the results is calculated. For this, we considered the number of dominant microorganisms, the structure of the flocs and their size, and all other observations resulting from the microscopical images together with the known operating conditions of the treatment plant.

Data were evaluated and compared with Origin Pro software for all three studied years: from 2019 to 2021. The comparison was individually made for all four main categories of species: bacteria, unicellular sessile, unicellular, and pluricellular along with their components.

## 3. Results

First, the physicochemical parameters are presented and discussed as follows below.

In general, the pH values are dependent on the impurities present in the wastewater, with a slight tendency to increase the alkaline value. In 2020 during the lockdown period (Figure 2), more preciously in April–June, the pH values are almost constant, but after the lockdown, in July-December, an increase in the pH values is observed. Also, in 2020 these values are shown to be lower than in the years 2019 and 2021, probably due to the pathogen reduction. The pH increases were temporally higher in the period from mid-March to early May.

In May–June, during the lockdown period, the chemical oxygen consumption (COD) (Figure 3) was shown to have lower values than in 2019 and 2021, as well as in September–December 2020. After the lockdown period (COD) showed improvement from late January/early February to mid-March.

During the lockdown period, in May–June, the biochemical consumption of oxygen BOD_5_ (Figure 4), was shown to have lower values than in 2019 and 2021, as well as in the months of September–December, after the lockdown period. Usually, the COD values are higher than the BOD_5_ values because all the organic compounds are oxidized, but our results show the opposite. The trend is decreasing BOD_5_ up to August when it reached the lowest value during the summer holiday period.

In 2020, (Figure 5), before the lockdown period, in January–February, TSS is shown to have higher values than 100 mg/L, even higher than in 2019 and 2021, after which the concentration decreases both in the lockdown period, as well as after the lockdown, up to the value of 60 mg/L in September, after which there was a slight increase, to 65 mg/L, but still did not reach the value of 100 mg/L.

An increase in ammonium concentration along the primary clarifiers (Figure 6), often signals septic conditions from excessive sludge build-up.

In April–July 2020, the values were lower than in 2019 and 2021. The dominant trend was the decrease in TN.

The months of May–June 2020 TP (Figure 7) present lower values than in 2019 and 2021. The lower values of nutrients, (Figure 7) the nitrogen and phosphorus content, show that the wastewater also requires the second stage of purification.

Average annual values for COD are usually higher than BOD_5_ results because all organic compounds are oxidized. In 2020, both COD and BOD_5_ had lower values than in 2019 or 2021, with higher oxygen consumption. The pH values do not show too much variation, probably the COVID-19 waste has a pH close to neutral pH. All these aspects are validated by Figure 8 which shows the average annual values for all the studied physicochemical paarameters: pH, COD (mg/L), BOD_5_ (mg/L), TSS (mg/L), TN (mg/L), and TP (mg/L) at primary decantation, in the years 2019, 2020, and 2021.

Depending on the quantity and quality of the aeration process and the quality of the organic substances in the wastewater (BOD_5_), a population of heterotrophic bacteria develops, which becomes food for *zooflagellates* and *ciliates*.

The stabilization of the organic material leads to the removal of *zooflagellates*. Since *rotifers* and *nematodes* have a longer life cycle (2 weeks or more), they only appear after the stabilization of the functioning of the biological stage.

Microorganisms need a source of energy, a source of carbon for the synthesis of new cellular material, as well as nitrogen TN and phosphorus TP for the synthesis of proteins, to support their reproduction process and functioning in optimal conditions.

The C:N:P ratio required for the culture of microorganisms involved in aerobic purification is 100:5:1, and for anaerobic is 200:5:1 [41].

With respect to biological composition, the analyzed activated sludge is made of bacteria, unicellular sessile, unicellular and pluricellular, as seen in Appendix A (which refers to comparative analysis regarding the species of microorganisms in activated sludge, identified in this study and those reported in the literature). According to the data presented in Appendix A, the following microorganisms presented in Table 1 were identified in our study, in the aeration tanks of the activated sludge.

Figure 9 shows microscopic images representative of the species of microorganisms identified in activated sludge from aeration tanks. The most representative species of microorganisms are: *Aspidisca cicada*, *Coleps*, *Litonotus*, *Rotifer*, *Vorticella convallaria*, *Ciliata free of the unknown*, and *Epistylis*.

Analyzing each group of microorganisms, significant variations appeared for half of the encountered bacteria (Figure 10)—except for the absent ones (*Sulphur bacteria*, *Sphaerotilus* spp., *Nocardia,* Type 021N, Type 1701, Type 0041).

The most abundant unicellular sessile organisms analyzed in the samples of the activated sludge were *Epistylis* spp. and *Vorticella convallaria* in all studied years, as Figure 11 emphasizes.

Regarding the pluricellular organisms (Figure 12), the *rotifers* were the most abundant in all three studied years (before and during the pandemic times) and reached the highest values of 2.0 in 2020, followed by 1.67 in 2021 and almost the same in 2019, with a value of 1.6.

Moving to the unicellular free microorganisms (Appendix A), which describes unicellular abundance in the three years of study: (a) 2019; (b) 2020; (c) 2021, a great difference exists between 2019, 2020, and 2021, respectively. There are quite a lot of *Tecamoeba*, *Amoeboids*. The differences between seasons are noticeable and only the temperature variation is obvious. As can be seen in all the figures, the number values are monthly averages for different types of microorganisms.

## 4. Discussion

Microscopic determinations were used to evaluate the diversity and dynamics of the microbial community from the activated sludge shown in Figure 9. The sludge from the Romanian WWTP had a structure that is characterized by a variety of species (as shown in Materials and Methods): bacteria, unicellular free and sessile, and pluricellular, out of which especially *ciliates* and *rotifers* were very common. Referring to bacteria, they are very important in the good treatment of wastewater and activated sludge from the WWTP, being strictly related to COD and BOD_5_. The presence of *Microtrix* (with an average of almost 2.0 over the three studied years, and peaks of 4.0 in 2021 and 3.0 in 2019 and 2020, respectively) and of *Spirilla*, *spirochetes* (with highest peaks of 4.0 in 2019 and 3.0 in 2020) is important due to their action of linking the small flocs from one place to another [42].

The most important microorganism indicator of good, stable and healthy activated sludge quality and wastewater treatment stability is the presence of *free swimming and stalked ciliates* along the *rotifers* [43]. It is known that *crawling ciliates* are playing an important role in the young to medium age activated sludge as they improve flocculation and remove loosely attached *bacteria* from the flocs [44]. The existence of the species *Aspidisca cicada* indicates that oxygenation is good in the aeration area of the tanks. *Coleps* spp. are not present here, because they are found only in low load stations [45], which is not our case. Even if *amphileptides*, *amoebe* and *tecamoebes* are present in the activated sludge, the quality of the biomass is not affected. The reason is the low presence of the three mentioned species (an average of only 0.4 for *amphileptides*, 0.38 for *amoebe* and 0.2 for *tecamoebes* for the three studied years). Overall, *protozoa* are one of the most common components in ecosystems and play an important role in wastewater purification processes. They are responsible for improving the quality of the effluent, by maintaining the density of dispersed bacterial populations [46].

Moving to unicellular sessile species, they are not represented by many individuals, most of them being inexistent (equals zero according to Figure 11). The presence of *Epistylis* spp. and their high value of 0.8 for 2019 and 2021, and 0.45 for 2020, indicate the well-functioning of the wastewater treatment plant, as well as the sufficient oxygenation, as Appendix A shows. The presence of *Vorticella* spp. in our activated sludge is correlated with a good system operation because these species tend to leave their stalks under bad treatment conditions such as low dissolved oxygen (DO) levels or toxicity [47]. *Vorticella convallaria* is relatively pretentious and thus is found only in wastewater treatment plants with good oxygenation, aspect valid for the studied WWTP and confirmed by the values of COD and BOD_5_ from Figure 3 and Figure 4, and Appendix A and the values of this species in the activated sludge (1.4 in 2019 and almost 1.0 in 2020 and 2021).

As for the pluricellular, the *Rotifers* are the most present or the only one present in the activated sludge. In 2020 they registered only positive values for the first semester, while in 2019 were the only pluricellular present the entire year along with two ups of *Nematodes*; for 2020, only the rotifers registered positive values. This is a sign of a very stable activated sludge environment [47]. The microscopic composition of the activated sludge collected from the studied WWTP is further detailed regarding each group of microorganisms and their influence and significance on the activated sludge and the wastewater treatment plant.

### 4.1. Bacteria

Over the years, no matter the absence or presence of the COVID-19 pandemic, some of the bacteria were not present, such as *Nocardia*, *Sphaerotilus* spp., type 021N, type 1701, and type 0041. Several morphologically particular filamentous microorganisms were observed. It was confirmed the presence of *Haliscomenobacter hydrossis*, *Thiothrix nivea*, *Sphaerotilus natans*, *Nostocoida limicola II*, *Eikelboom* Type 1851, members of the *Eikelboom* Type 021N group II, based on hybridization with their corresponding oligonucleotide determinations. It was noticed a higher filament diversity in industrial compared to municipal plants [42]. More likely, they play a role in the occurrence of specific filaments.

In Figure 3 the curve trend varied in a very interesting way from 2019 to 2020 and then to 2021. In 2019 when everything was still normal, the daily routine was known for years, except for the *Free bacteria* that encountered ups and downs in the first half of the year, the rest of the bacteria had almost the same trend. Another exception was made by the *Zoogloea* spp. which was quite linear in distribution. A common phenomenon happened during August and September when all fell to zero (perhaps because of the summer holiday period and temporary inactivity of most industries, knowing the negative effect of the industrial wastewater discharges [48,49]). Moving on to 2020, the SARS-CoV-2 pandemic and lockdown made their mark. During the state of emergency, most of the bacteria registered constancy in the variation. For some of the bacteria, this trend extended until autumn, when in fact all fell to 0. *Spirilla*, *Spirochetes* were those out of the pattern in the first three semesters of 2019. An interesting aspect appeared in 2021, when all bacteria that presented variations during the year, had ups and downs in the same months of the year, with peaks in March, June, and September.

*Spirilla*, *Spirochetes* registered the highest values of four in 2019 and 2021, before the pandemic and after the lockdown. In 2020, the maximum value was lower, 3.5 in February and May. In comparison, the peaks of the other variables were considerably lower (with 0.5, even 1 point—a significant difference). An exception appeared for 2019 when the two of them joined *Tokophrya* spp. with a small increase of 0.5 in July.

### 4.2. Unicellular Sessile

If we discuss the variation trend, the unicellular sessile organisms presented almost the same pattern as bacteria: peaks followed by constancy in 2019, two peaks and constant variation in 2020, and only ups in 2021 (discussing about the organisms’ variation encountered during the study). The presence of the two-oxygen demanding most abundant unicellular sessile organisms emphasized a good treatment process of the activated sludge from the WWTP both before and during the pandemic times. The absence of *Votircella microstoma* indicates a good level of oxygenation. Moreover, if we compare Figure 10 and Figure 11, the microorganisms—bacteria and unicellular sessile—presented the same variation over the years, even over the seasons.

### 4.3. Unicellular Free

In Appendix A, Resistance, *Coleps* spp., *Euplotes* spp., *Aspidisca lynceus*, and *Glaucoma* spp. were zero in all three years. All the rest of the free unicellular varied both as fluctuation and limits. *Amoeboids* and *Paramecium* spp. kept the variation limits (around 2.5) in 2019 and 2020, even if the curve trend in 2020 is with multiple ups and downs compared to 2019 when their variation was more stable. *Amoeboids* may predominate in the MLSS floc during start-up periods of the activated sludge process or when the process is recovering from an upset condition [43,50].

*Flagellate* predominance may be associated with a light-dispersed MLSS floc, a low population of bacteria, and a high organic load (BOD), which is not true in our case. As denser sludge flocs develop, the flagellate predominance will decrease with an increase in bacteria. When the flagellates no longer can successfully compete for the available food supply, their population decreases to the point of insignificance [34,43,50] Overall, because of the COVID-19 pandemic period in 2020, that imposed temporary shutoffs and repeated opening and closing of different important industries, services, education, and commerce, all free microorganisms determined in the activated sludge of the WWTP had brutal ups and downs during the year. By comparing the graphs for 2019 and 2021, it is easy to conclude that the economy and life started to recover in 2021, an aspect demonstrated by the 2021 limits’ variation of some important unicellular organisms (like: *Amoeboids*, *Amphileptus*, *Paramecium* spp., *Zooflagelate*) that approaches to those for 2019. A comparative analysis regarding the species of microorganisms in activated sludge, identified in this study, and those reported in the literature [46,48,49,51,52,53,54,55] is also shown in Appendix A.

Related to the WWTP, the relative predominance of ciliates is an indicator of process stability. This predominance is associated with the efficiency of treatment under various loading conditions. An increase or decrease in the predominance of these organisms may be indicative of processes being upset before there is a major effect on the process.

### 4.4. Pluricellular

Regarding our results shown in Figure 5, *Rotifers* are more abundant at higher cell residence times and are an indication of a high level of treatment of the wastewater. *Fungi* (low pH), *Nematods*, and Insects were also detected, but their abundance was lower than *rotifers*, even absent in the case of the three of them in 2021, respectively, in the case of *Fungi* in 2019. Therefore, their presence or absence did not affect the quality of the wastewater treatment.

Overall, the predominance of *ciliates* and *rotifers* in the MLSS and their variation in abundance is a sign of good sludge quality [47]. Treatment under these conditions, with the proper return, waste activated sludge, and aeration rates, can be expected to produce a high-quality effluent [47]. Inversely, a predominance of filamentous organisms (bacteria) and a limited number of ciliates is characteristic of poor-quality activated sludge. This condition is commonly associated with a sludge that is not the case of the studied wastewater treatment plant. The sludge flocs are usually light and fluffy because it has a low density [56]. Thus, the activated sludge from the studied WWTP is of good and healthy quality and the wastewater treatment process returns a high-quality effluent.

Microorganisms from the activated sludge—bacteria, unicellular free and sessile, and pluricellular—fluctuated differently from 2019 to 2021. Variations were noticed in 2019, except for the summer holiday that coincides with the months of August and September. In 2020, for quite the same types of species as in 2019, the variations increased over months, with lots of ups and downs compared to 2019 (when life before COVID-19 was normal). This change in the variation pattern is also strongly related to the physicochemical parameters: COD, BOD_5_, pH, TSS, TN, and TP. For 2021, when life and the economy started to recover from the COVID-19 pandemic, microorganism abundance presented peaks in March, June, and September, according to the activity of the reborn industries, services, and other facilities. The exception was made by the unicellular free species; they presented the highest variation in 2021, followed by 2020 and 2019, respectively. This means that we are discussing a healthy and stable activated sludge, along with a good purifying wastewater treatment process.

## 5. Conclusions

Through this study we have shown that the physicochemical parameters and biological composition of the activated sludge from the studied wastewater treatment plant situated in a metropolitan area of Romania were significantly influenced by the COVID-19 pandemic. Following our experiment, we noticed that in the long-term study, analyses concerning the physicochemical parameters varied somehow as expected, according to the lifestyle type of each year: 2019, 2020, and 2021. Before the COVID-19 pandemic, TSS, BOD_5,_ and COD presented the highest values, followed by the ones from 2021 when industries and other sectors started to revive. The reason is the reintroduction of different agents and substances into the wastewater. BOD_5_ and COD had higher values (an average of 220 for COD and 92.5 for BOD_5_) that indicated a higher oxygen consumption in all three years. The year 2020, the full pandemic year, presented the lowest values of all physicochemical parameters (due to the absence of contaminants from the industrial waters). The only exception is made by the pH which registered the same average value of 7.52 in all the three studied years, showing that the COVID-19 waste had a pH close to neutral.

The biological aspects of the activated sludge of the wastewater treatment plant performed between 2019 and 2021 emphasized that in all situations, microorganisms such as bacteria, unicellular free and sessile, and pluricellular species are present. The most important microorganism indicator for good sludge and purifying treatment process is the presence of *protozoa* (*ciliates*, *flagellates*) and the *rotifers* in great notes of abundance. By eating the bacteria, the protozoa helped to provide a clear effluent.

The microbiological state of the activated sludge significantly differed before and during the COVID-19 pandemic, an aspect strengthened by the significant variation of the physicochemical parameters. Of all the microorganisms, the *protozoa* and *rotifers* were the most abundant in all three years. The *rotifers* reached the highest value of 2.0 in 2020, while before the pandemic, in 2019, their value was 1.6 and towards the end of the pandemic, in 2021, the value started to decrease again to 1.67 (notes of abundance). Overall, the normal life from 2019 before COVID-19 transformed into a capping during the state of emergency and lockdown from 2020. This demonstrates that no matter if the industry is working or not, if people are staying home and the wastewater is most domestic, the WWTP can successfully face any situation and deliver clean effluents. The same conclusion appeared for 2021 when the industry started to recover. The most representative taxa of microorganisms for the studied activated sludge are: *Aspidisca cicada*, *Coleps*, *Litonotus*, *Rotifer*, *Vorticella convallaria*, *Ciliata* free of the unknown, and *Epistylis*.

Therefore, we also emphasized that in general, the same types of microorganisms are in the activated sludge from the wastewater treatment plant, no matter the absence or existence of COVID-19 or the trend of work (workplace or teleworking). The only difference appeared in the variation of the presence of microorganisms according to the change of the physicochemical parameters (pH, COD, BOD_5_, TSS, TN, TP). In all cases, the treatment process was good, providing a clear effluent. Considering all of these, further research could be made by extending the investigation to a greater number of WWTPs and to various locations to see whether differences between regions and countries exist or not.

## Figures and Tables

**Figure 1 ijerph-19-11323-f001:**
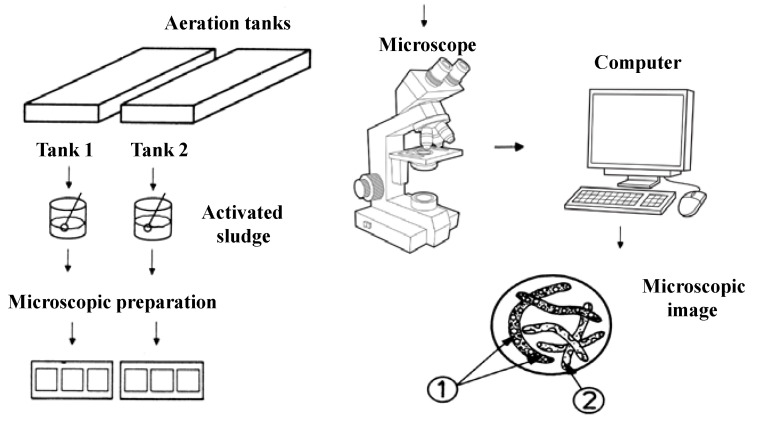
Schematic representation of the microscopic characterization. Microscopic images 1 and 2: different microorganisms in the activated sludge.

**Figure 2 ijerph-19-11323-f002:**
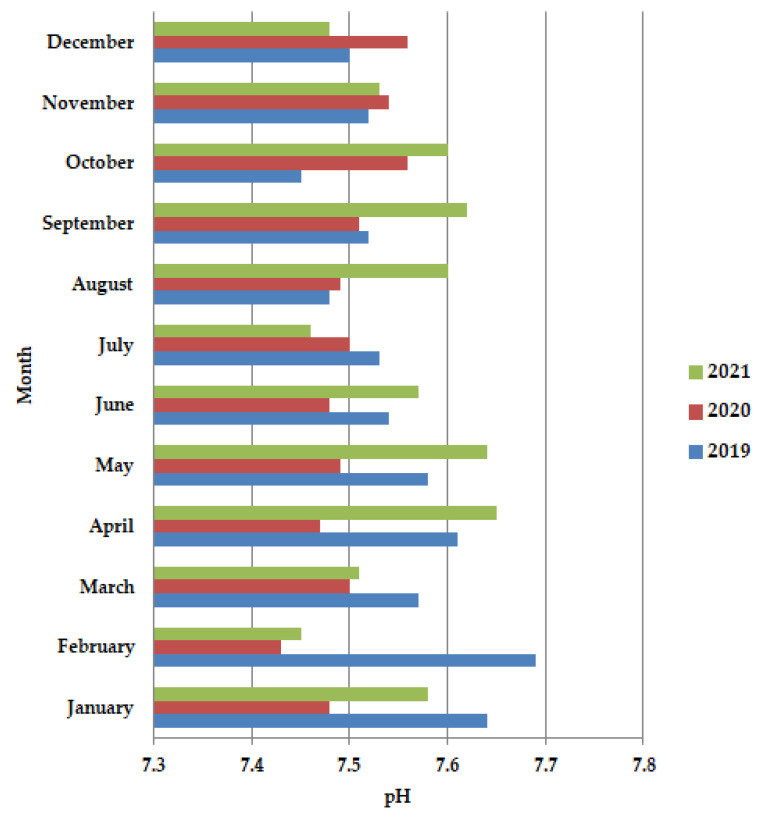
Average monthly pH values, at primary decantation, in the years 2019, 2020, and 2021.

**Figure 3 ijerph-19-11323-f003:**
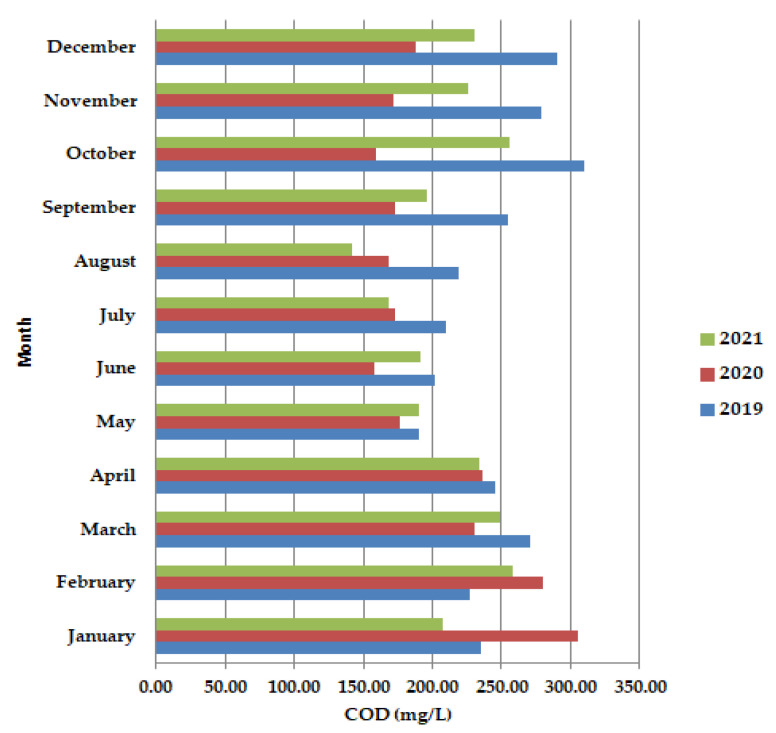
Average monthly COD (mg/L) values, at primary decantation, in the years 2019, 2020, and 2021.

**Figure 4 ijerph-19-11323-f004:**
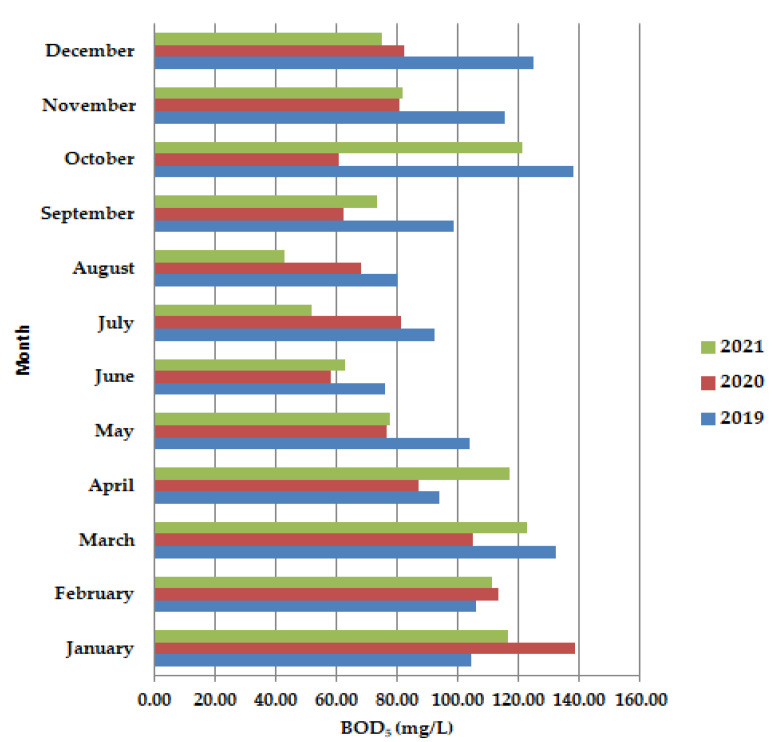
Average monthly BOD_5_ (mg/L) values, at primary decantation, in the years 2019, 2020, and 2021.

**Figure 5 ijerph-19-11323-f005:**
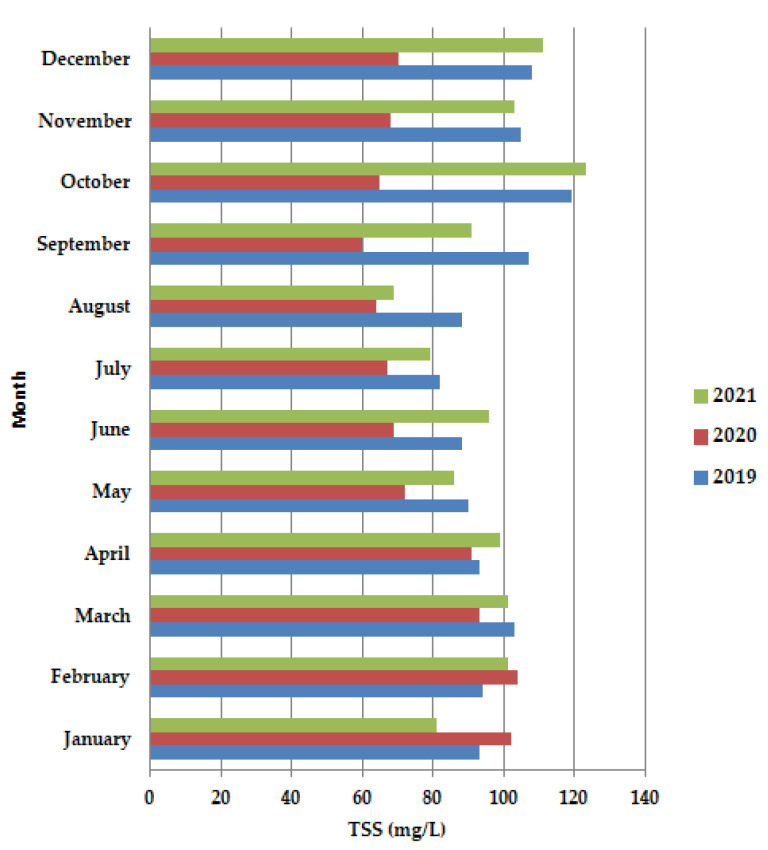
Average monthly TSS (mg/L) values, at primary decantation, in the years 2019, 2020 and 2021.

**Figure 6 ijerph-19-11323-f006:**
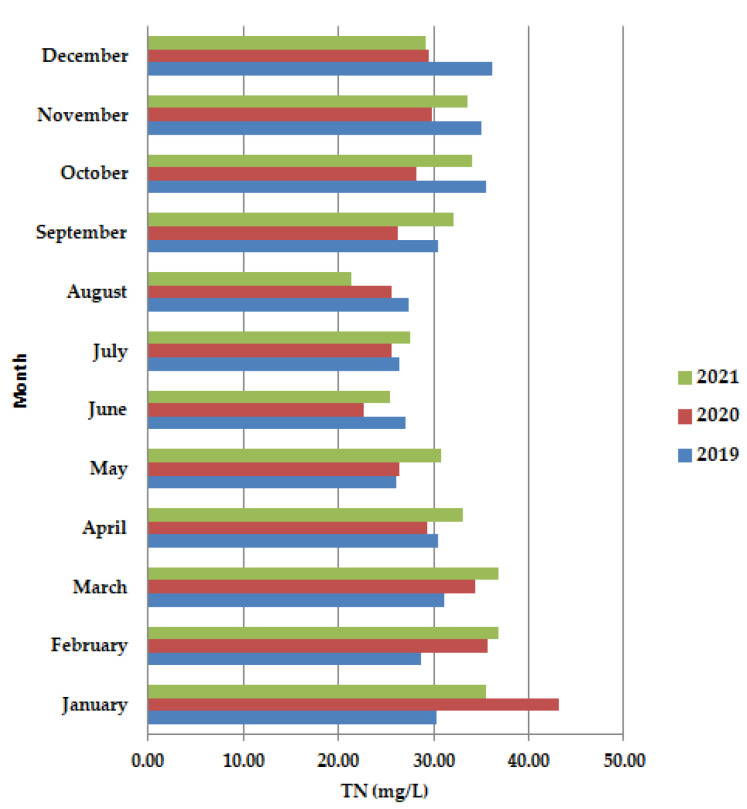
Average monthly TN (mg/L) values, at primary decantation, in the years 2019, 2020, and 2021.

**Figure 7 ijerph-19-11323-f007:**
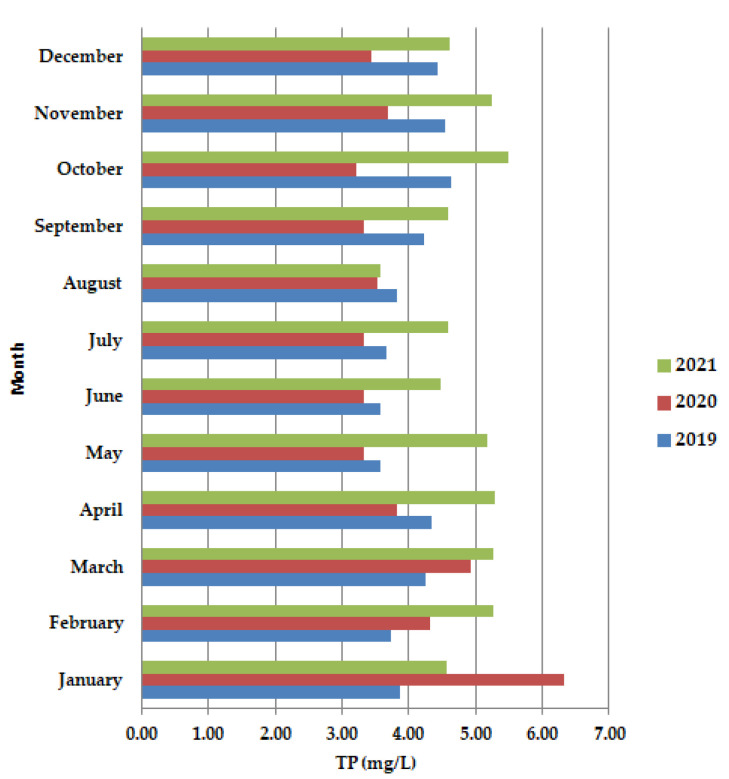
Average monthly TP (mg/L) values, at primary decantation, in the years 2019, 2020, and 2021.

**Figure 8 ijerph-19-11323-f008:**
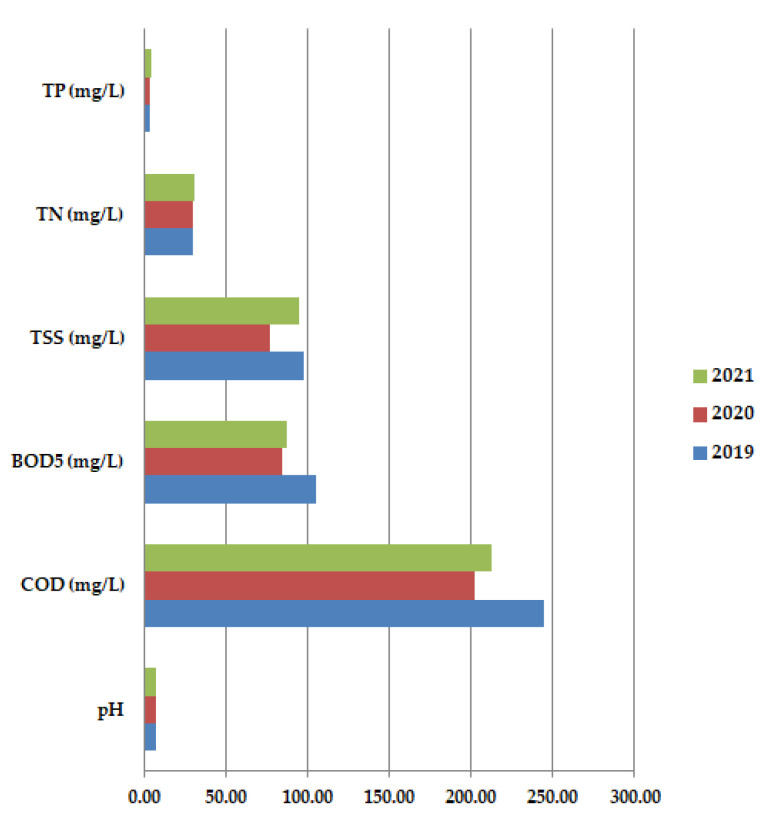
Average annual values for pH, COD (mg/L), BOD_5_ (mg/L), TSS (mg/L), TN (mg/L) and TP (mg/L) at primary decantation, in the years 2019, 2020 and 2021.

**Figure 9 ijerph-19-11323-f009:**
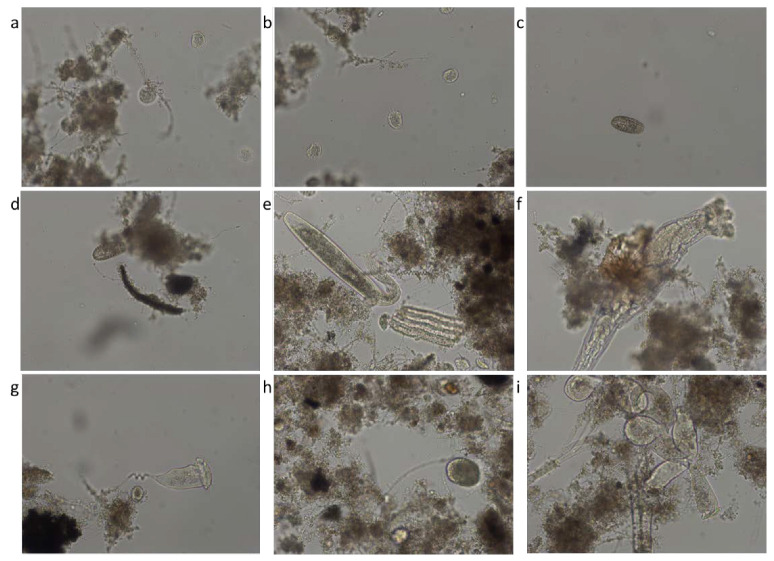
Identified species: (**a**) *Aspidisca cicada*; (**b**) *Aspidisca cicada*; (**c**) *Coleps*; (**d**) *Coleps*; (**e**) *Litonotus*; (**f**) *Rotifer*; (**g**) *Vorticella convallaria*; (**h**) *Ciliata free of the unknown*; (**i**) *Epistylis*.

**Figure 10 ijerph-19-11323-f010:**
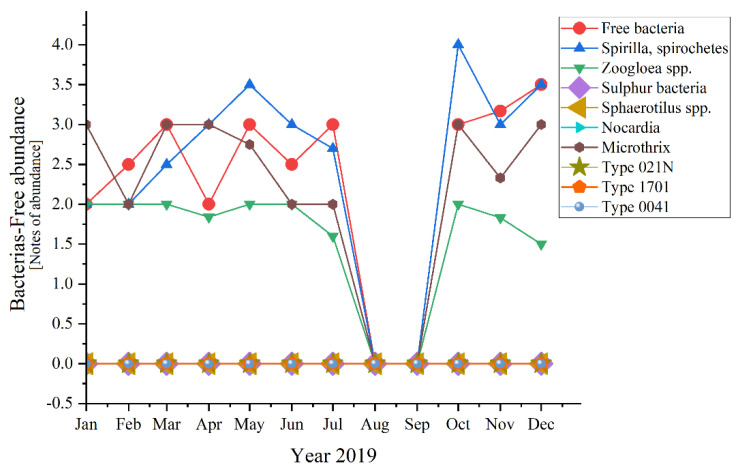
Bacterial abundance in the three years of study: 2019; 2020; 2021.

**Figure 11 ijerph-19-11323-f011:**
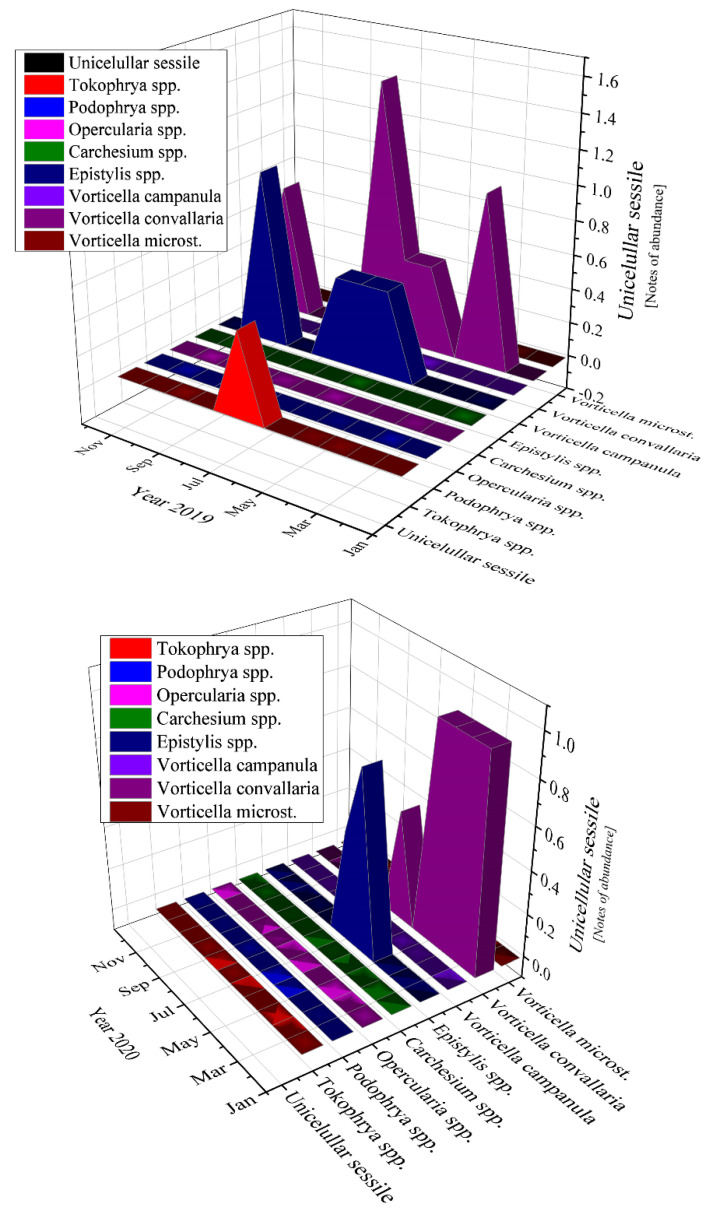
Unicellular sessile abundance in the three years of study: 2019; 2020; 2021.

**Figure 12 ijerph-19-11323-f012:**
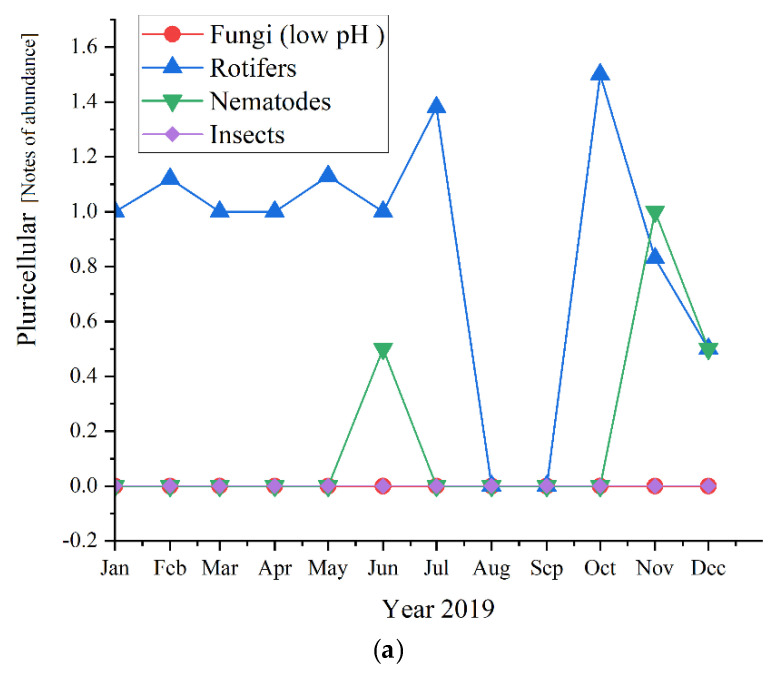
Pluricellular abundance in the three years of study: (**a**) 2019; (**b**) 2020; (**c**) 2021.

**Table 1 ijerph-19-11323-t001:** Groups and types of microorganisms found in the study.

Group of Microorganisms	Types of Microorganisms	Group of Microorganisms	Types of Microorganisms
1.Bacteria	Free bacteria	3.Unicellular	*Coleps* spp.
	*Spirilla*, *Spirochetes*		*Euplotes* spp.
	*Zoogloea* spp.		*Aspidisca lynceus*
	*Sulphur bacteria*		*Aspidisca cicada*
	*Sphaerotilus* spp.		*Chilodonella* spp.
	*Nocardia*		*Litonotus* spp.
	*Microthrix*		*Amphileptus*
	Type 021N		*Tecamoeba*
	Type 1701		*Amoeboids*
	Type 0041		*Paramecium* spp.
			*Dexiostoma campyla*
2.Unicellular	*Tokophrya* spp.		*Glaucoma* spp.
sessile	*Podophrya* spp.		*Zooflagelate*
	*Opercularia* spp.		*Flagelattes*
	*Carchesium* spp.		*Wandering cells*
	*Epistylis* spp.		*Resistance*
	*Vorticella campanula*		
	*Vorticella convallaria*	4.Pluricellular	*Fungi*
	*Vorticella microst*.		*Rotifers*
			*Nematodes*
			Insects

## Data Availability

Not applicable.

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
