# Peer review of "Biological Activated Sludge from Wastewater Treatment Plant before and during the COVID-19 Pandemic"

_ijerph, 2022, doi:10.3390/ijerph191811323_

Round 1

Reviewer 1 Report

1. The summary is very general, with no specific research results.

2. According to the title and the introduction the Authors intended to present the microbiological state of the activated sludge before and after the Covid-19 pandemic. There is no clearly defined aim of the research, e.g. in the end of the introduction.

3. In the Materials and Methods chapter, there is no description of the biological part of the sewage treatment plant - whether it is a conventional activated sludge system or whether they are biological reactors (anaerobic, anoxic or oxygenic zones) – it is very important. Therefore, it is not known what specific part of the biological system has been subjected to microbiological analyzes.

4. Chapter 2 (3rd paragraph) describes the sampling for testing is unclear. It is impossible to know how and how many of these attempts were actually taken. No information on the repeatability of the tests.

5. On page 5 of the article (second paragraph), it is stated that the specification of microorganisms from the tested activated sludge is included in Table 1 - it is not in the text of Table 1.

6. Figures - diagrams are unreadable.

7. There are no units on the ‘y’ axis - no reference to determine the size / value of the data provided.

8. There are no units on the y axis - no reference to determine the size / value of the data provided.

9. Interpretation of results, Chapter 4, is illegibly presented. There is no clear division into before and after Covid-19. The description provided is chaotic, disordered. The reviewer realizes that the interpretation of such results is not easy. Therefore, the Authors should additionally present the results in a tabular, graphical, selective manner, so that a reader who has not performed these studies and does not know their details, can read the main trend and research results.

10. The first paragraph of the Conclusions section refers to the Introduction section.

If the authors decide to summarize the article and the conclusions at the same time, this chapter should be titled: Summary and Conclusions.

The conclusions should also include measurable results (resulting from the diagrams).

11. In the article, the Authors often mention: Supplementary Figure / Table - what does this description refer to?

Author Response

Dear Editors and Reviewers,

We would like to thank you for your suggestions and for your significant contribution to improving this paper. Your comments are of great significance to our manuscript. We are deeply grateful for your help and hope that the correction will meet with your approval.

All changes are track changed in the paper and below we wrote point-to-point responses to reviewers` comments.

Reviewer 2 Report

The manuscript “ijerph-1864230” deals with the surveillance and monitoring of activated sludge process during Covid-19 regime. The manuscript is severely lacking in terms of design, data collection, findings presentation as well as supporting information. Following are my comments for improvement of the manuscript.

1)     Introduction section! The relevant literature is significantly lacking. The researchers are providing comprehensive details on the type of microorganisms and their growth phases which is meaningless here. In addition, the researchers are even concluding that protozoa and rotifers are major responsible microorganisms responsible for smooth run of activated sludge process. Somewhere in between of introduction section, the researchers indicate their study objectives about surveillance of WWTPs of Romania during Covid-19 in the years 2019-2021, however, rearranging and placing relevant literature in line with current research theme is mandatory requirement for fulfilling the purpose of current section.

2)     Methodology section! The researchers are claiming that activated sludge process worked very well and therefore, they checked activated sludge for various microscopic analysis. In such a case, it’s mandatory requirement to present results of WWTPs indicating parameters including BOD, COD, TSS, TVSS, TKN, TP, EPS and SMP etc. The methodology section should provide indicating of monitoring of such parameters along with detailed analysis procedure.

3)     Results, Discussion section! The researchers just indicate different organisms associated with biological sludge; at least Covid-19 regime year wise description must have been separately discussed. For instance, if rotifers are in high quantity in certain time or season, the researchers must indicate the reason. Overall, on one side, they are saying that rotifers and ciliates are mandatory requirement for good treatment while on other side they are claiming if such microbes are not found in dominance even then treatment process is not affected due to other types of microorganisms present in biological sludge. Again in such a case it would have been recommended to provide details on treatment parameters in order to correlate with different types of microbes.

4)     Line 251-253! Why is it so that small amphileptides, amoebe and tecamoebes do not decrease the quality of the biomass?

5)     Results, discussion section can’t be improved if researchers don’t have time series data of different WWTPs critical parameters, and in such case, manuscript can’t be improved in line with my comments. Same goes with abstract and conclusion section as well.

6)     Overall, the manuscript lacks in terms of novelty as well as data presentation.

Author Response

(The authors gave the same response as above.)

Round 2

Reviewer 2 Report

The manuscript has been significantly improved and now warrants publication in journal. From my comments, I hope that besides having in depth knowledge, researchers have now gain expertise on how to present manuscript appropriately in future as well while working on biological wastewater treatment systems.